# Elevated Serum Interleukin-8 Level Correlates with Cancer-Related Cachexia and Sarcopenia: An Indicator for Pancreatic Cancer Outcomes

**DOI:** 10.3390/jcm7120502

**Published:** 2018-12-01

**Authors:** Ya-Chin Hou, Chih-Jung Wang, Ying-Jui Chao, Hao-Yun Chen, Hao-Chen Wang, Hui-Ling Tung, Jung-Ting Lin, Yan-Shen Shan

**Affiliations:** 1Institute of Clinical Medicine, College of Medicine, National Cheng Kung University, Tainan 704, Taiwan; yachi2016@yahoo.com.tw (Y.-C.H.); Poemcage@gmail.com (C.-J.W.); surgeon.chao@gmail.com (Y.-J.C.); tp6cl4mp62@yahoo.com.tw (H.-Y.C.); esr2wang@gmail.com (H.-C.W.); 2Department of Clinical Medical Research, National Cheng Kung University Hospital, College of Medicine, National Cheng Kung University, Tainan 704, Taiwan; 3Division of General Surgery, Department of Surgery, National Cheng Kung University Hospital, College of Medicine, National Cheng Kung University, Tainan 704, Taiwan; ritaiap.tw@gmail.com; 4School of Medicine, College of Medicine, National Cheng Kung University, Tainan 704, Taiwan; billin9769@hotmail.com

**Keywords:** pancreatic cancer, cancer cachexia, sarcopenia, interleukin (IL)-1β, IL-6, IL-8, tumor necrosis factor (TNF)-α

## Abstract

Cancer cachexia (CC), characterized by body weight loss and sarcopenia, contributes to over 20% of all cancer-related death. Approximately 80% of pancreatic cancer (PC) patients develop CC during disease progression. Pro-inflammatory cytokines, including interleukin (IL)-1β, IL-6, IL-8, and tumor necrosis factor (TNF)-α, have been correlated with CC; however, its prognostic significance remains unclear. In this study, serum levels of the CC-related cytokines were determined in normal donors and PC patients. IL-8 expression was assessed in PC tissue microarrays. The correlation of levels of each cytokine with disease progression, weight loss, and sarcopenia was calculated. The relationships among the baseline variables, CC, and IL-8 expression with disease progression were examined using univariate and multivariate analyses. Of these mentioned cytokines, only serum IL-8 level was elevated in the locally advanced group (*n* = 55) compared with the normal (*n* = 17) and resected groups (*n* = 55). Serum IL-8 level was positively correlated with CC status, weight loss, sarcopenia, but was negatively correlated with total psoas area (TPA). IL-8 expression in tissue samples was also positively associated with weight loss. Furthermore, serum IL-8 level was an independent predictor of survival. In conclusion, elevated serum IL-8 level significantly correlates with CC and sarcopenia and can be used as a prognostic indicator in PC.

## 1. Introduction

Cancer cachexia (CC), a syndrome defined by unintended weight loss greater than 5% in over 6 months or sarcopenia with any degree of weight loss greater than 2%, is a major cause of death in more than 20% of all cancer patients [1,2]. The main symptoms of CC include reduced body weight, skeletal muscle loss with or without the loss of adipose tissue, weakness, and anorexia, which worsen survival outcomes, impair the quality of life, decrease the response to therapy, increase the toxicity of anticancer therapy, and reduce physical activity [3]. The therapeutic options for this multi-factorial and complex syndrome so far remain limited.

It has been reported that patients with pancreatic cancer (PC) have the highest incidence of CC and experience severe symptoms of this syndrome [4]. Approximately 80% of PC patients develop CC during the disease course, and up to 30% die from CC-related complications [5]. Tumor removal is probably the best way to cure CC [6]; however, more than 80% of PC patients have unresectable disease and should be treated with systemic chemotherapy, the main method for prolonging survival although its effectiveness is limited [4,7]. In addition, CC can be exacerbated by systemic chemotherapy and, thus, reduce patients’ tolerance to anti-cancer agents [8]. Accordingly, understanding CC etiology in PC patients is urgent for the development of intervention strategies to alleviate this condition.

Current treatment strategies for pancreatic CC remain limited and based on supportive care practices. A variety of drugs have been used to inhibit the production and release of pro-inflammatory cytokines including interleukin (IL)-1β, IL-6, and tumor necrosis factor (TNF)-α and thus contribute to appetite stimulation and weight gain [5]. Moreover, IL-8 is a pro-inflammatory cytokine and a promising predictor for prognosis of PC [9]. Recent research has indicated that CC is associated with pro-inflammatory signals from tumor cells, systemic inflammation in the host, and widespread metabolic changes [10]. Our unpublished data found that IL-8 level was elevated in the conditioned media from PC cells co-cultured with tumor associated macrophages or pancreatic stellate cells and promoted glycolysis in PC cells. Although several pro-inflammatory cytokines, including IL-1β, IL-6, IL-8, and TNF-α, have been implicated in the clinical features of CC in patients with gastrointestinal or non-small cell lung cancer [11,12], whether detecting these cytokines is effective in predicting the prognosis of patients with CC remains uncertain. Given that PC patients have the highest frequency of weight loss than those with other common cancers [13], our primary objective in this study was to explore the association of CC status with the serum IL-1β, IL-6, IL-8, or TNF-α levels in patients with resected or locally advanced PC. The secondary objective was to evaluate the prognostic utility of IL-8 protein expression alone and in combination with CC status in PC.

## 2. Materials and Methods

### 2.1. Study Population and Clinical Parameters

Data from all PC patients between 2001 and 2012 were reviewed using electronic medical records. Patients were followed from their initial visit to medical oncology before receiving any cancer treatment until death or 2012. The extent of body weight loss was compared to usual body weight at the time of diagnosis of PC and related symptoms were recorded. The use of human specimens was approved by the Institutional Review Board of National Cheng Kung University Hospital (Tainan, Taiwan). A total of 146 patients were included in this study and were divided into 2 groups based on the extent of the disease: resected (*n* = 91) or locally advanced/unresectable (*n* = 55). The defined criterion according to a previous report [1] for cachexia was weight loss greater than 5%, or weight loss greater than 2% in individuals already showing depletion according to current bodyweight and height (body-mass index (BMI) <20 kg/m^2^) or skeletal muscle mass depletion (sarcopenia). Weight loss was defined as losing at least 5% of initial body weight and maintaining the loss for at least 6 months. CA19-9 level of 100 U/mL was used as the cut-off thresholds according to the review concluding that an elevated CA 19-9 serum levels (>100 U/mL) is associated with a poor prognosis [14]. Details of the study cohort assembly are presented in Figure 1.

### 2.2. Serum Sample Collection and Cytokine Detection Assays

Serum was separated from peripheral blood samples that were collected by venipuncture (10 mL; BD Vacutainer glass serum tube) and was stored at −80 °C in aliquots. Serum IL-1β, IL-6, IL-8, and TNF-α levels of 55 patients in the resected group and 55 patients in the locally advanced group were measured using an enzyme-linked immunosorbent assay (ELISA; R&D, HS600B, HS800, SSLB00C, and HSTA00D, Quantikine, Minneapolis, MN, USA) according to the manufacturer’s instructions. All samples were analyzed in duplicate.

### 2.3. Computed Tomography (CT) Image Analysis

All computed tomography (CT) images were taken for diagnostic purposes and viewed on standard desktop computer screens using Pictures Archiving and Communication System (PACS, version 3.0.10.2 (BM2), INFINITT Healthcare, Seoul, Korea). Muscle mass was estimated by measuring the cross-sectional area of the psoas muscle at the third lumbar vertebrae using a PACS measuring ruler. The values were normalized for the patient height (total psoas area (TPA) mm^2^/m^2^). Sarcopenia, or muscle mass depletion, was defined by gender-specific cut-off points of <385 mm^2^/m^2^ for female patients and <545 mm^2^/m^2^ for male patients [15].

### 2.4. Immunofluorescence Staining and Measurement

Paraffin-embedded tissue microarrays (TMAs) used to survey IL-8 expression is composed of 91 tumor resection specimens of pancreatic cancer that include 10 adjacent normal/cancer pairs. Adjacent normal and cancer sites in donor paraffin blocks were identified by an experienced pathologist using matching hematoxylin and eosin reference slides, and the TMAs were constructed in duplicate or quadruplicate cores, each 2 mm in diameter. TMA sections were stained with anti-human IL-8 antibody (1:500 dilution; Abcam, ab7747, Cambridge, UK) followed by incubation with the appropriate fluorescently conjugated secondary antibody (1:1000 dilution; Abcam, ab150080, Cambridge, UK). Cell nuclei were counterstained with 4’,6-diamidino-2-phenylindole (DAPI; Molecular Probes, D3571, Eugene, OR, USA). The full-field fluorescence image of each core was acquired using the FACS-like Tissue Cytometry system (Tissue Gnostics, Vienna, Austria). The percentage of positive cells in each image was further quantified by TissueQuest system and the software TissueFAX (Tissue Gnostics, Vienna, Austria). Tumor heterogeneity was evaluated using Pearson’s correlation coefficient (*r*) in two different cores from the same tumor blocks as described previously [16].

### 2.5. Statistical Analysis

Statistical analyses were performed using the Graphpad Prism 5.0 and SPSS Statistics 17.0 (IBM, Endicott, NY, USA) software. The mean value of IL-8 expression in the tumor or serum samples was used to classify patients into the high-expression group (≥mean) or the low-expression group (<mean). Variables in the univariate analysis (*p* < 0.1) were then entered into Cox proportional hazards models. Multivariate analysis for overall survival (OS) or disease-free survival (DFS) was conducted using the Cox proportional hazard model to calculate the hazard ratios (HRs) and corresponding 95% confidence intervals (CI). Data were expressed as the means± standard error in all experiments. Significance was set at *p* < 0.05.

## 3. Results

### 3.1. Cancer Cachexia (CC) Status Is Associated with Poor Clinical Outcome in Pancreatic Cancer (PC) Patients

One hundred and ten patients with CC had significantly shorter OS (*p* = 0.0002) and DFS (*p* = 0.018) than did those without CC (Figure 2A,B). CC was present in 64.8% of the resected cases, in contrast to 92.7% of the locally advanced cases (*p* = 0.000). In the resected group, CC had a significant negative impact on survival (*p* = 0.001) and was associated with tumor grade (*p* = 0.011), weight loss (*p* = 0.000), and fatigue (*p* = 0.025). For patients with locally advanced disease, CC was significantly related to survival (*p* = 0.000), weight loss (*p* = 0.000), BMI (*p* = 0.000), fatigue (*p* = 0.006), and anorexia (*p* = 0.008; Table 1).

### 3.2. Interleukin (IL-8) Is an Applicable Diagnostic Indicator for CC Status

In this study, we determined the clinical relevance of serum IL-1β, IL-6, IL-8, and TNF-α levels in PC patients with CC. As shown in Figure 3, IL-1β, IL-6, IL-8, and TNF-α serum levels were significantly higher in the locally advanced group than in the other groups (*p* < 0.0001). However, only IL-8 levels were found to be increased in serum of the resected group compared with the normal group (*p* < 0.05; Figure 3C). Moreover, there were no significant differences in serum IL-1β or TNF-α levels between patients with and without CC (Figure 4A–C,J–L). Of all cases and the resected group, but not the locally advanced group, serum IL-6 levels were increased in patients with CC (*p* = 0.0006, *p* = 0.0206, and *p* = 0.1563, respectively; Figure 4D–F). Figure 4G,H,I show that, of all cases, the resected group, and locally advanced group, serum IL-8 levels were significantly elevated in patients with CC than in those without CC (*p* < 0.0001, *p* = 0.0453, and *p* = 0.0131, respectively).

### 3.3. High Serum IL-8 Level Commonly Occurs in Patients Exhibiting Weight Loss and Muscle Wasting

Weight loss has been recognized as a formal definition of cachexia; therefore, we evaluated the correlation between serum cytokines and weight loss. In all samples, serum levels of IL-6 and IL-8, but not IL-1β and TNF-α, were positively correlated with weight loss (*r* = 0.2824, *p* = 0.0028; *r* = 0.2254, *p* = 0.0179; *r* = 0.0899, *p* = 0.3505; *r* = 0.1058, *p* = 0.2715, respectively; Figure 5D,G,A,J). In the resected group, both serum IL-6 and IL-8 levels were positively correlated with weight loss (*r* = 0.4865, *p* = 0.0002; *r* = 0.3065, *p* = 0.0229, respectively; Figure 5E,H). In the locally advanced group, only serum IL-8 but not IL-6 level was positively correlated with weight loss (*r* = 0.2800, *p* = 0.0384; *r* = 0.2400, *p* = 0.0776, respectively; Figure 5I,F). However, neither serum IL-1β nor TNF-α level was correlated with weight loss in the resected group and the locally advanced group (Figure 5B,C,K,L).

Sarcopenia, or muscle mass depletion, is a common issue in cancer patients and is an independent predictor of immobility and mortality [15]. Therefore, we determined the clinical relevance of these serum cytokines in muscle wasting during disease progression. There were no statistically differences in serum IL-1β or TNF-α levels between patients with and without sarcopenia (Figure 6A–C,J–L) or no relationship between serum IL-1β or TNF-α levels and TPA (Figure 7A–C,J–L). Increased serum IL-6 levels were only found in sarcopenic patients of the locally advanced group but not all samples and the resected group (*p* = 0.0488, *p* = 0.0970, and *p* = 0.4438, respectively; Figure 6F,D,E). IL-6 was negatively correlated with TPA in the locally advanced group but not in all cases and the resected group (*r* = −0.3019, *p* = 0.0350; *r* = −0.1298, *p* = 0.2151; *r* = −0.0199, *p* = 0.8980, respectively; Figure 7F,D,E). Only IL-8 was significantly upregulated in serum of sarcopenic patients of all cases, the resected group, or the locally advanced group (*p* = 0.0131, *p* = 0.0348, and *p* = 0.0177, respectively; Figure 6G,H,I) and negatively correlated with total psoas area (TPA) (*r* = −0.2544, *p* = 0.0139 for all cases, *r* = −0.4211, *p* = 0.0044 for resected group, and *r* = −0.3736, *p* = 0.0082 for locally advanced group, respectively; Figure 7G,H,I).

### 3.4. IL-8 Expression Are Related to CC Symptoms and Disease Progression in PC

To substantiate the potential of IL-8 as a CC biomarker, we determined IL-8 expression in TMA sections or serum samples. Figure 8A shows IL-8 expression in pancreatic normal or tumor tissues. IL-8 expression was significantly higher in cancer lesions than in their normal counterparts (*p* = 0.0002) or adjacent normal tissues (*p* < 0.0001; Figure 8B,C). High IL-8 expression in tumor tissues was also associated with tumor size (*p* = 0.045), CA19-9 levels (*p* = 0.031), and anorexia (*p* = 0.004; Table 2). In the resected group, IL-8 expression in tumor tissues was significantly higher in patients with CC than in those without CC (*p* = 0.0146; Figure 8D) and was positively correlated with weight loss (*r* = 0.2317, *p* = 0.0271; Figure 8E). Importantly, IL-8 levels in tumor specimens were consistent with that in serum from the same patients (*r* = 0.2777, *p* = 0.0401; Figure 8F). High IL-8 expression in tumor tissues was significantly associated with shorter DFS but not OS compared with low IL-8 expression (*p* = 0.037 and *p* = 0.0467, respectively; Figure 9A,B). By contrast, high serum IL-8 levels showed a significant correlation with poor OS and DFS (*p* = 0.029 and *p* = 0.038, respectively; Figure 9C,D).

### 3.5. The Combination of IL-8 with CC Predicts Poor Prognosis of PC

Because IL-8 levels were associated with CC status, we classified the patients into 3 groups to determine the contribution of IL-8 and CC to clinical outcomes: group 1 (patients without CC and with low IL-8 expression), group 2 (patients without CC and with high IL-8 expression or patients with CC and low IL-8 expression), and group 3 (patients with CC and high IL-8 expression). Group 3 had significantly worse survival than the other groups (*p* = 0.034 for tumor tissues; *p* = 0.009 for all serum samples; Figure 9E,F, respectively). Multivariate analysis show that CC was an independent predictor for OS (hazard ratio, HR, 18.629; 95% CI, 5.390 to 64.382; *p* = 0.000) or DFS (HR, 1.712; 95% CI, 1.092 to 2.685; *p* = 0.019). Weight loss was identified as an independent factor for OS (HR, 7.682; 95% CI, 2.617 to 22.544; *p* = 0.000) or DFS (HR, 1.581; 95% CI, 1.027 to 2.433; *p* = 0.038). Anorexia was associated with an increased risk of death (HR, 2.260; 95% CI, 1.359 to 3.757; *p* = 0.002). Patients with high IL-8 expression in tumor tissues had shorter DFS (HR, 1.752; 95% CI, 1.080 to 2.843; *p* = 0.023). This was also observed in patients with high serum IL-8 levels (HR, 1.615; 95% CI, 1.045 to 2.495; *p* = 0.031 for OS and HR, 1.539; 95% CI, 1.021 to 2.319; *p* = 0.040 for DFS). High IL-8 expression in combination with CC was an independent predictor of OS (HR, 2.555; 95% CI, 1.042 to 6.261; *p* = 0.040 for tumor tissues; HR, 3.840; 95% CI, 1.508 to 9.777; *p* = 0.005 for all serum samples) and DFS (HR, 1.996; 95% CI, 1.088 to 3.954; *p* = 0.047 for tumor tissues; HR, 1.928; 95% CI, 0.990 to 3.755; *p* = 0.045 for all serum samples). The results are summarized in Table 3.

## 4. Discussion

CC, one of the most common cancer-related complications, has a profound adverse impact on patients’ quality and length of life [17]. This syndrome develops progressively through various stages: precachexia, cachexia, and lastly refractory cachexia [1]. Late-stage cachexia in advanced cancer patients is refractory to all treatments [18]. Early detection for CC would improve treatment effect; unfortunately, little is known concerning effective biomarkers for CC. In this study, higher serum levels of IL-1β, IL-6, IL-8, and TNF-α were observed in cancer patients of the locally advanced group than those of the normal and resected groups. However, only IL-8 was upregulated gradually with disease progression. Compared to other cytokines, serum IL-8 level was increased in patients with CC or sarcopenia, positively correlated with weight loss, but negatively correlated with TPA and survival among all groups. Similarly, IL-8 expression was also elevated in tumor tissues of PC patients with CC compared with those without CC and positively correlated with weight loss. Therefore, serum IL-8 level may improve the accuracy in predicting pancreatic cancer outcomes.

The occurrence of CC in cancer patients depends on the host response to tumor progression, including activation of the inflammatory response and energetic inefficiency involving the mitochondria [18]. Evidence has revealed that CC results from the spillover effects of cytokine production by tumors on individual organs, leading to anorexia, muscle wasting, fatigue, loss of adipose tissue, and increased lipolysis involving several signaling pathways and metabolites [19]. Recently, high serum levels of IL-1β, IL-6, IL-8, and TNF-α in cancer patients have been linked to CC pathogenesis [12,20,21]. IL-6, IL-8, and TNF-α, but not IL-1β, are upregulated in prostate carcinoma (CaP) patients with CC compared with those without CC. Notably, IL-8 level was higher in advanced CaP patients than in healthy volunteers and patients with organ-confined CaP. Although both IL-6 and TNF-α levels were also increased in advanced CaP patients or advanced PC patients with CC, no statistically significant differences were identified [22,23,24]. Increased IL-6, IL-8, and TNF-α level were also found in gastrointestinal cancer patients with cachexia. CC patients showed higher value of IL-8 (55.6-fold) but not IL-6 (9.8-fold) and TNF-α (1.6-fold) when compared with the control and with weight-stable cancer group [21]. An agreement was made in low-third gastric cancer that serum IL-8 level was significantly increased in cancer patients compared with that in controls and was further up-regulated in patients with CC than in those without CC [25]. Consistent with these reports, we observed that only IL-8 level was significantly elevated in the resected group compared with that in the normal group and was further up-regulated in the locally advanced group than in the resected group. We also found that IL-8 levels in both the resected and locally advanced groups were positively correlated with CC status and linked to weight loss.

Recent comprehensive research using several animal models has demonstrated that IL-6 plays a role in muscle wasting [26,27,28]. Although clinical studies attempt with IL-6 or its receptor antibodies have had some success ameliorating lung cancer-related anemia and weight loss, these treatments could not prevent tumor expansion and its effect on muscle mass remains unknown [29,30,31]. We show that serum IL-6 levels were increased in sarcopenic patients in the locally advanced group but not the resected group. In contrast, increased IL-8 levels were associated with sarcopenia status but negatively correlated with TPA in both the resected and the locally advanced groups. Meanwhile, IL-1β and TNF-α serum levels did not correlate with sarcopenia status. Therefore, IL-8 may constitute a major proponent in the pathogenesis of sarcopenia.

The cachectic state is mainly problematic in cancer and typifies poor prognosis rather than the tumor burden [32]; however, tumor burden affects the loss of skeletal muscle, a facet of CC. For instance, colorectal cancer patients display reduced postprandial muscle protein synthesis and a trend toward increased muscle protein breakdown, but tumor resection reverses these derangements [33]. Interestingly, numerous studies in human cancers, including cancers of the lung [34], liver [35], stomach [36], and pancreas [9], have reported that IL-8 expression was increased in serum or tissue specimens from cancer patients and was correlated with tumor size. IL-8 concentration was correlated with the number of IL-8-producing tumor cells in vitro, and serum IL-8 level rapidly dropped after surgical excision in vivo [37]. These findings indicate an accurate correlation between the IL-8 level and tumor burden. In this study, IL-8 expression in PC tissue specimens was positively correlated with tumor size, suggesting that IL-8 could be an excellent factor to define the cachectic condition.

Of the 198 PC patients who underwent resection, 39.9% presented with CC and had significantly worse survival than did those without CC [38]. Here, we found that CC was presented in 64.8% of the resected cases and 92.7% of the locally advanced cases. Patients with CC had significantly shorter OS and DFS than those without CC. Both serum and tissue IL-8 levels were correlated with PC patient survival. Taken together, these results suggest that IL-8 is a particularly critical factor not only in PC progression but also in CC development.

In summary, our study provided the first compelling evidence that high IL-8 level represents the most prominent characteristic of CC than IL-1β, IL-6, and TNF-α levels in both resected and locally advanced PC patients. It is necessary to measure serum IL-8 level in order to assess the prognosis of PC patients. Further investigations are required to clarify the biological mechanisms to evaluate whether IL-8 inhibition in pancreatic cancer serves as a promising therapeutic strategy for attenuating CC and has the potential to retard disease progression.

## Figures and Tables

**Figure 1 jcm-07-00502-f001:**
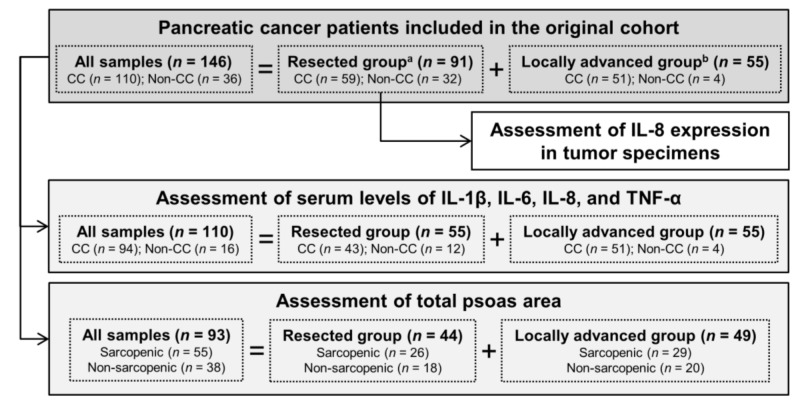
Assembly of study cohort. a, patients with surgical treatment. b, patients with unresected disease. Abbreviations: CC, positive cancer cachexia status; Non-CC, negative cancer cachexia status; IL, interleukin; TNF, tumor necrosis factor.

**Figure 2 jcm-07-00502-f002:**
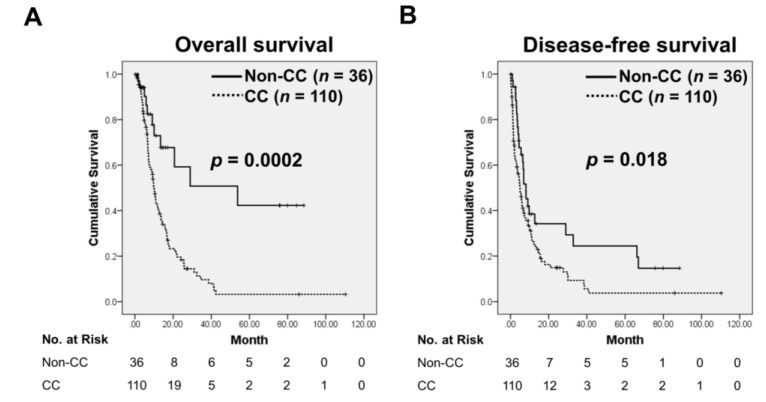
Kaplan–Meier curves of overall survival (OS) (**A**) and disease-free survival (DFS) (**B**) in relation to CC status at diagnosis among patients with pancreatic cancer (PC). CC, positive cancer cachexia status; Non-CC, negative cancer cachexia status.

**Figure 3 jcm-07-00502-f003:**
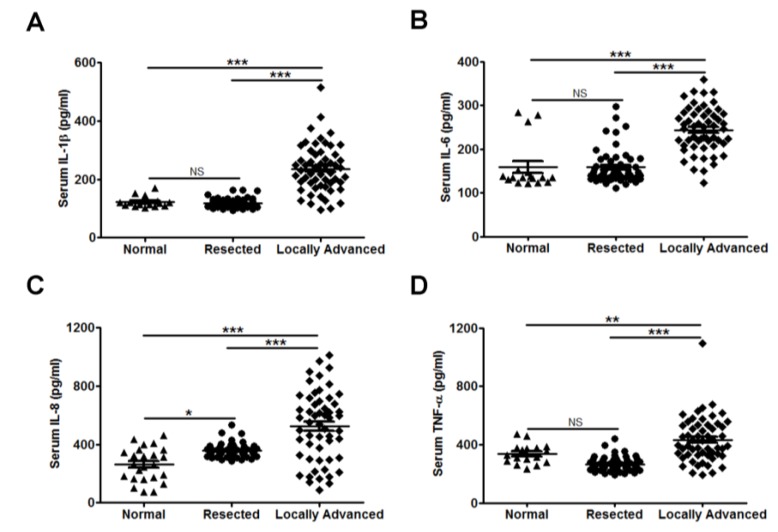
Serum IL-1β, IL-6, IL-8, and TNF-α levels in healthy donors and PC patients. IL-1β (**A**), IL-6 (**B**), IL-8 (**C**), and TNF-α (**D**) concentrations were measured by ELISA. Values represent mean ± standard error of the mean; NS, not significant; *, *p* < 0.05; **, *p* < 0.01; ***, *p* < 0.001 compared with the normal group, as determined using one-way analysis of variance (ANOVA).

**Figure 4 jcm-07-00502-f004:**
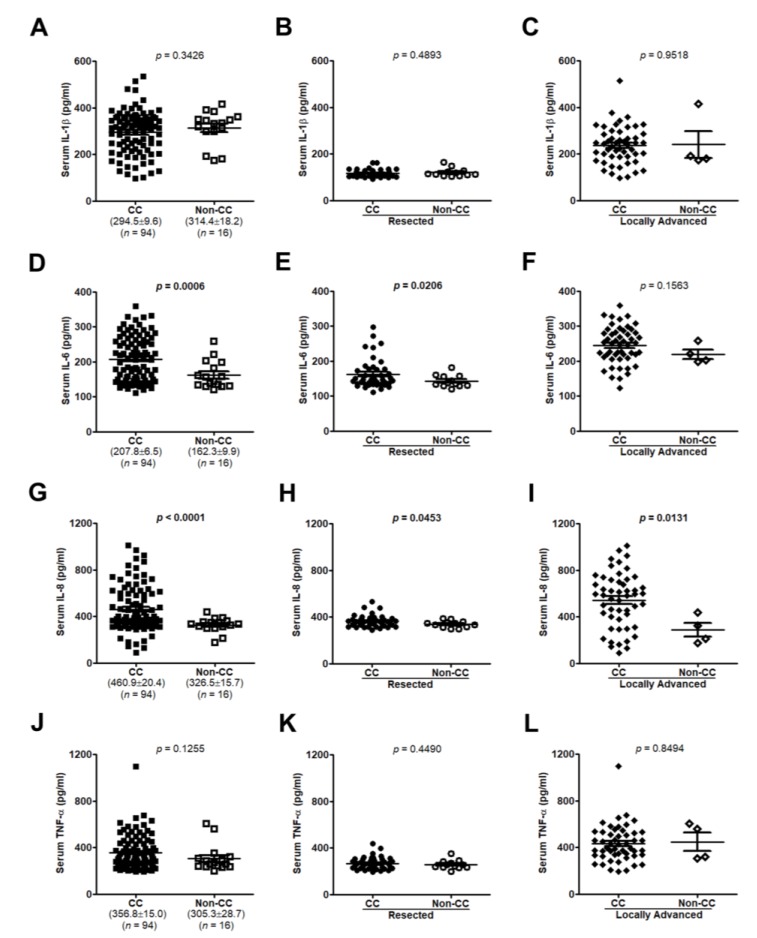
Serum IL-1β, IL-6, IL-8, and TNF-α levels in PC patients with or without CC. IL-1β, IL-6, IL-8, and TNF-α concentrations in patients with or without CC from all samples (**A**,**D**,**G**,**J**), the resected group (**B**,**E**,**H**,**K**), or the locally advanced group (**C**,**F**,**I**,**L**) were measured by enzyme-linked immunosorbent assay (ELISA). Values represent mean ± standard error of mean; two-tailed Student’s *t*-test. CC, patients with cancer cachexia; Non-CC, patients without cancer cachexia.

**Figure 5 jcm-07-00502-f005:**
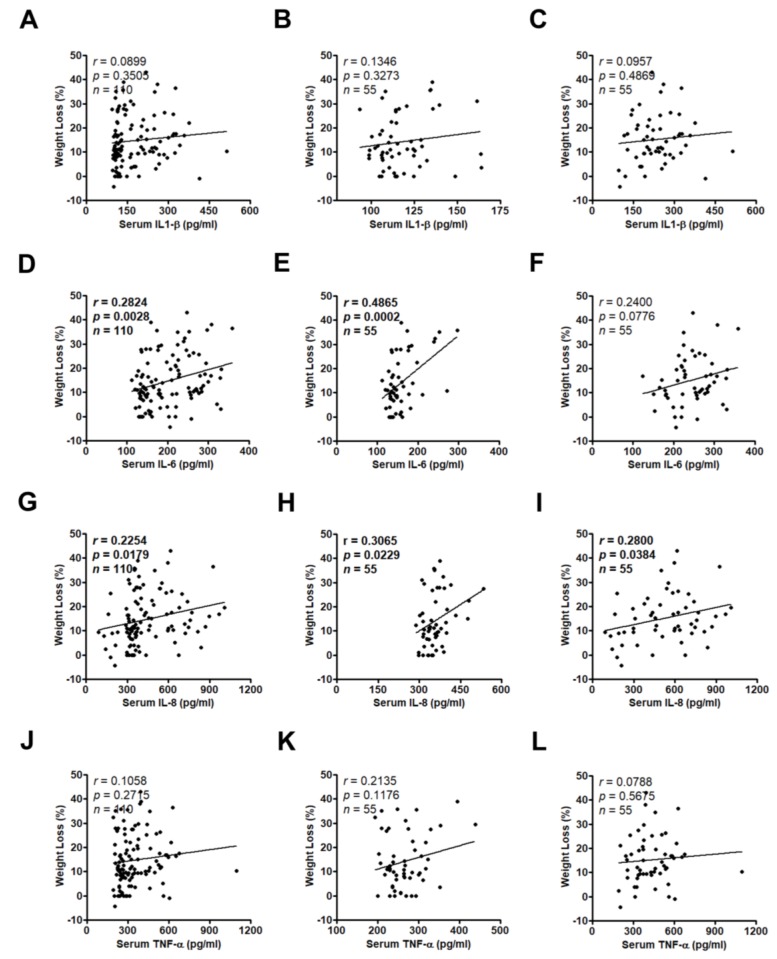
The correlation between IL-1β, IL-6, IL-8, or TNF-α serum concentrations from all patients (**A**,**D**,**G**,**J**), the resected group (**B**,**E**,**H**,**K**), or the locally advanced group (**C**,**F**,**I**,**L**) and weight loss was analyzed by Pearson’s correlation analysis. The bold value indicates *p* < 0.05.

**Figure 6 jcm-07-00502-f006:**
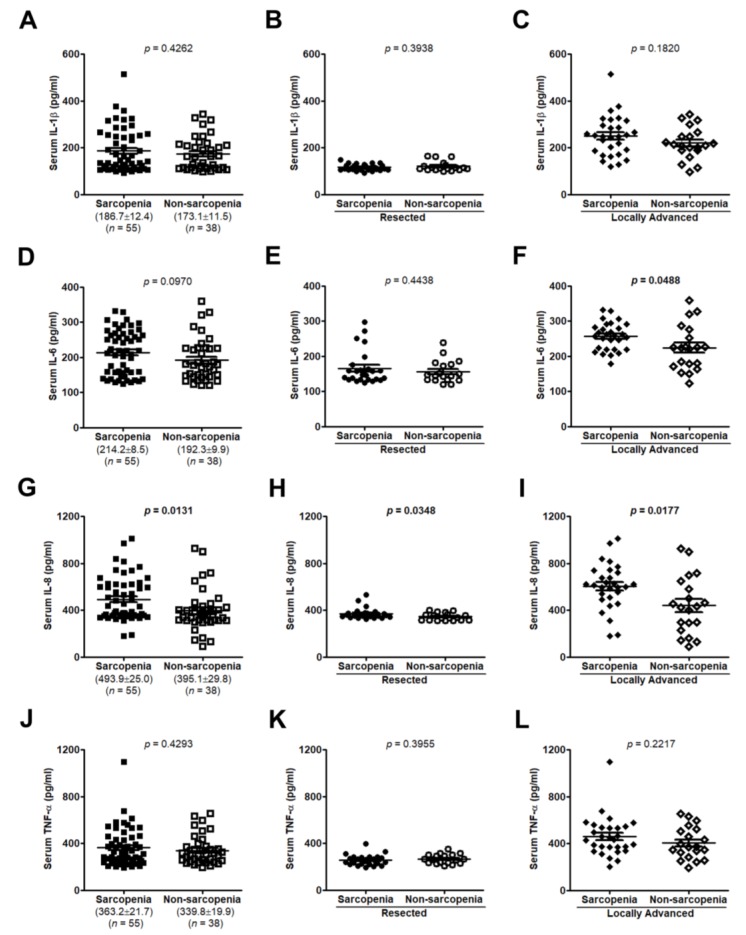
Serum IL-1β, IL-6, IL-8, and TNF-α levels in PC patients according to sarcopenia status. IL-1β, IL-6, IL-8, and TNF-α concentrations in patients with or without sarcopenia from all samples (**A**,**D**,**G**,**J**), the resected group (**B**,**E**,**H**,**K**), or the locally advanced group (**C**,**F**,**I**,**L**) were measured by ELISA. Values represent mean ± standard error of the mean; two-tailed Student’s *t*-test. Sarcopenia, patients with sarcopenia; Non-sarcopenia, patients without sarcopenia.

**Figure 7 jcm-07-00502-f007:**
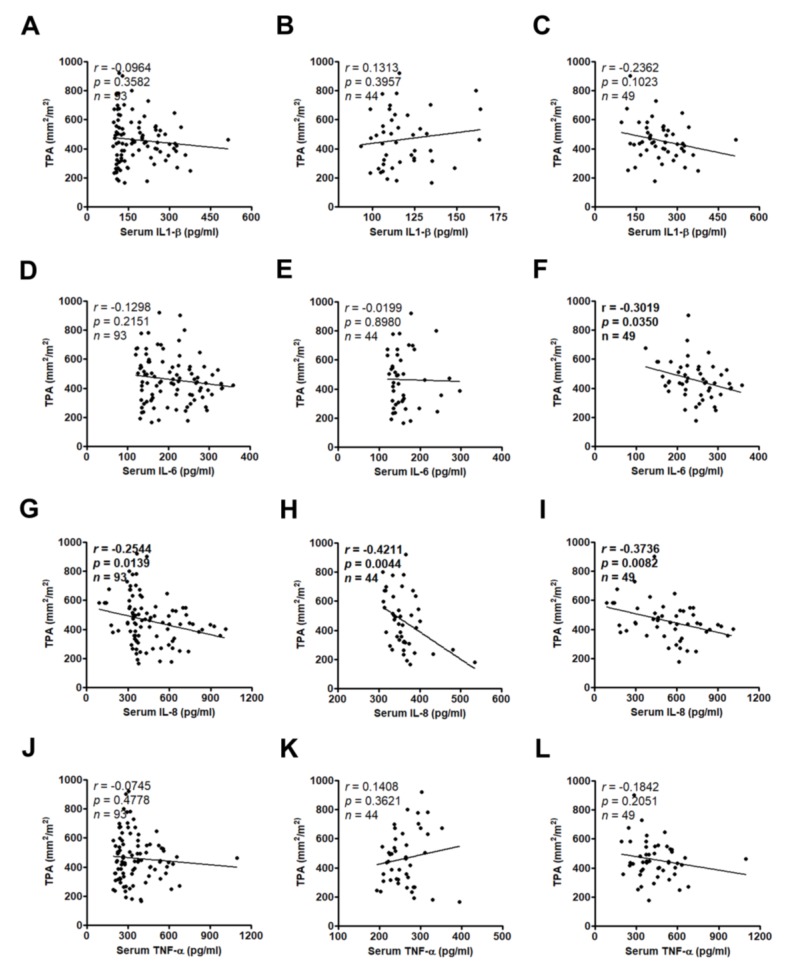
The correlation between IL-1β, IL-6, IL-8, or TNF-α serum concentration from all patients (**A**,**D**,**G**,**J**), the resected group (**B**,**E**,**H**,**K**), or the locally advanced group (**C**,**F**,**I**,**L**) and total psoas area was analyzed by Pearson’s correlation analysis. The bold value indicates *p* < 0.05. TPA, total psoas area.

**Figure 8 jcm-07-00502-f008:**
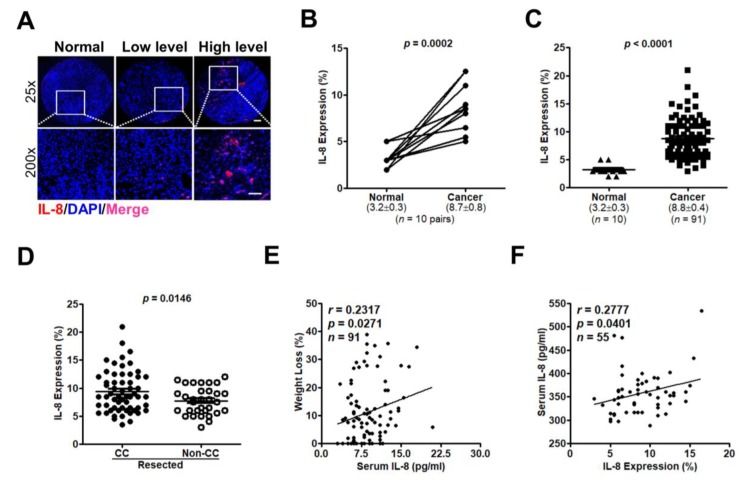
Representative immunofluorescence images showing IL-8 expression in normal and tumor samples. (**A**) Tissue microarray cores are shown at ×25 original magnification (Bar = 200 µm). The area in the white rectangle is shown as a magnified view below the image (Bar = 50 µm; ×200 magnification). The paraffin-embedded patient tissue samples were stained with anti-IL-8 antibody (red) and DAPI (4′,6-diamidino-2-phenylindole) for cell nuclei (blue). (**B**) IL-8 expression was compared between PC specimens and their normal counterparts. (**C**) IL-8 expression levels (mean ± standard error of the mean) were compared between normal and PC tissues. (**D**) PC patients with or without CC showed different levels of IL-8 in tumor tissues. (**E**) Pearson’s correlation analysis demonstrated that IL-8 expression level in tumor tissues was positively correlated with weight loss. (**F**) The correlation between IL-8 expression in tumor specimens and IL-8 levels in serum samples was analyzed by Pearson’s correlation analysis.

**Figure 9 jcm-07-00502-f009:**
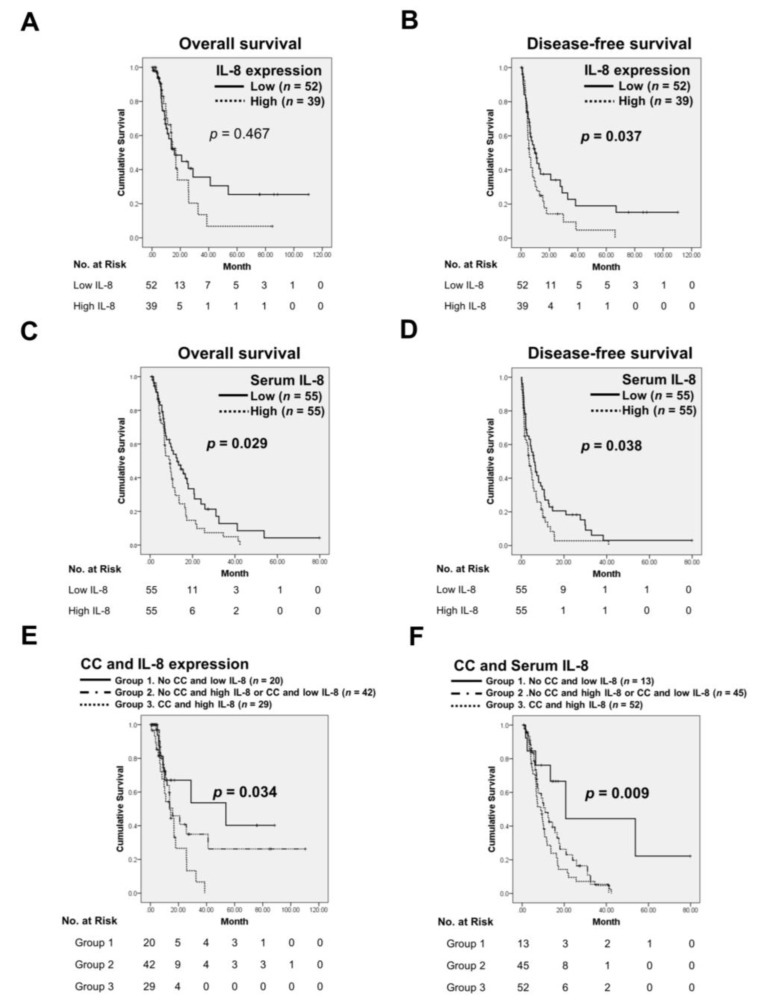
Kaplan–Meier plots for OS and DFS according to the IL-8 expression level in tumor tissues (**A**,**B**) or serum samples (**C**,**D**). The co-presence of high IL-8 expression in tumor tissues (**E**) or serum samples (**F**) and CC status had a positive correlation with OS in PC patients by Kaplan–Meier analysis. CC, positive cancer cachexia status; Non-CC, negative cancer cachexia status.

**Table 1 jcm-07-00502-t001:** Clinical characteristics of PC patients with or without CC.

Characteristic	Resected	Locally Advanced	*p*-Value	Resected			Locally Advanced		
				Non-CC	CC	*p*-Value	Non-CC	CC	*p*-Value
**Number of Patients**	91	55		32	59		4	51	
**Age, Median (Range)**	66 (37–85)	67 (41–87)							
**Sex**									
Male	59 (64.8)	31 (56.4)	0.308	20 (62.5)	39 (66.1)	0.731	3 (75.0)	21 (41.2)	0.189
Female	32 (35.2)	24 (43.6)		12 (37.5)	20 (33.9)		1 (25.0)	30 (58.8)	
**Tumor Location**									
Head	56 (61.5)	27 (49.1)	**0.002**	16 (50.0)	40 (67.8)	0.415	1 (25.0)	26 (51.0)	0.502
Neck	6 (6.6)	4 (7.3)		3 (9.4)	3 (5.1)		1 (25.0)	3 (5.9)	
Body/tail	16 (17.6)	16 (29.1)		7 (21.9)	9 (15.3)		1 (25.0)	15 (29.4)	
Uncinate process	13 (14.3)	2 (3.6)		6 (18.8)	7 (11.9)		0 (0.0)	2 (3.9)	
unknown	0 (0.0)	6 (10.9)		0 (0.0)	0 (0.0)		1 (25.0)	5 (9.8)	
**Tumor Size**									
<3 cm	34 (37.4)			12 (37.5)	22 (37.3)	0.984			
≥3 cm	57 (62.6)			20 (62.5)	37 (62.7)				
**Lymph Nodes**									
Negative	46 (50.5)			15 (46.9)	31 (52.5)	0.606			
Positive	45 (49.5)			17 (53.1)	28 (47.5)				
**Margin Status**									
R0	65 (71.4)			23 (71.9)	42 (71.2)	0.184			
R1	22 (24.2)			6 (18.8)	16 (27.1)				
R2	4 (4.4)			3 (9.4)	1 (1.7)				
**Tumor Grade**									
Poorly diff.	17 (18.7)			1 (3.1)	16 (27.1)	**0.011**			
Moderate diff.	48 (52.7)			18 (56.2)	30 (50.8)				
Well diff.	26 (28.6)			13 (40.6)	13 (22.0)				
**Stage**									
I	11 (12.1)			5 (15.6)	6 (10.2)	0.814			
II	74 (81.3)			25 (78.1)	49 (83.1)				
III	4 (4.4)			1 (3.1)	3 (5.1)				
IV	2 (2.2)			1 (3.1)	1 (1.7)				
**CA19-9**									
<100 U/mL	37 (40.7)	12 (21.8)	**0.019**	12 (37.5)	25 (42.4)	0.651	1 (25.0)	11 (21.6)	0.873
>100 U/mL	54 (59.3)	43 (78.2)		20 (62.5)	34 (57.6)		3 (75.0)	40 (78.4)	
**Adjuvant Therapy**									
Yes	36 (39.6)	45 (81.8)	**0.000**	14 (43.8)	22 (37.3)	0.547	2 (50.0)	43 (84.3)	0.087
No	55 (60.4)	10 (18.2)		18 (56.2)	37 (62.7)		2 (50.0)	8 (15.7)	
**Diabetes Mellitus**									
Yes	63 (69.2)	32 (58.2)	0.175	9 (28.1)	19 (32.2)	0.687	1 (25.0)	22 (43.1)	0.479
No	28 (30.8)	23 (41.8)		23 (71.9)	40 (67.8)		3 (75.0)	29 (56.9)	
**Survival**									
**Yes**	47 (51.6)	1 (1.8)	**0.000**	24 (75.0)	23 (39.0)	**0.001**	1 (25.0)	0 (0.0)	**0.000**
**No**	44 (48.4)	54 (98.2)		8 (25.0)	36 (61.0)		3 (75.0)	51 (100.0)	
**Cancer Cachexia**									
Yes	59 (64.8)	51 (92.7)	**0.000**						
No	32 (35.2)	4 (7.3)							
**Weight Loss**									
<5%	32 (35.2)	8 (14.5)	**0.007**	32 (100.0)	0 (0.0)	**0.000**	4 (100.0)	4 (7.8)	**0.000**
>5%	59 (64.8)	47 (85.5)		0 (0.0)	59 (100.0)		0 (0.0)	47 (92.2)	
**BMI**									
≤20 kg/m^2^	53 (58.2)	33 (60.0)	0.542	18 (56.2)	35 (60.3)	0.623	0 (0.0)	33 (64.7)	**0.000**
>20 kg/m^2^	32 (35.2)	21 (38.2)		13 (40.6)	19 (32.8)		3 (75.0)	18 (35.3)	
Unknown	5 (5.6)	1 (1.8)		1 (3.1)	4 (6.9)		1 (25.0)	0 (0.0)	
**Fatigue**									
Yes	37 (40.7)	35 (63.6)	**0.007**	8 (25.0)	29 (49.2)	**0.025**	0 (0.0)	35 (68.6)	**0.006**
No	54 (59.3)	20 (36.4)		24 (75.0)	30 (50.8)		4 (100.0)	16 (31.4)	
**Anorexia**									
Yes	38 (41.8)	34 (61.8)	**0.019**	9 (28.1)	29 (49.2)	0.052	0 (0.0)	34 (66.7)	**0.008**
No	53 (58.2)	21 (38.2)		23 (71.9)	30 (50.8)		4 (100.0)	17 (33.3)	
**CRP**									
<5 mg/L	16 (17.6)	6 (10.9)	0.208	8 (25.0)	8 (13.6)	0.374	0 (0.0)	6 (11.8)	0.548
>5 mg/L	71 (78.0)	43 (78.2)		23 (71.9)	48 (81.4)		4 (100.0)	39 (76.5)	
Unknown	4 (4.4)	6 (10.9)		1 (3.1)	3 (5.1)		0 (0.0)	6 (11.8)	
**Hemoglobin**									
<12 g/dL	36 (39.6)	24 (43.6)	0.149	12 (37.5)	24 (40.7)	0.767	2 (50.0)	22 (43.1)	0.903
>12 g/dL	55 (60.4)	29 (52.7)		20 (62.5)	35 (59.3)		2 (50.0)	27 (52.9)	
Unknown	0 (0.0)	2 (3.6)		0 (0.0)	0 (0.0)		0 (0.0)	2 (3.9)	
**Albumin**									
<3.2 g/dL	18 (19.8)	11 (20.0)	0.060	8 (25.0)	10 (16.9)	0.515	2 (50.0)	9 (17.6)	0.273
>3.2 g/dL	72 (79.1)	39 (70.9)		24 (75.0)	48 (81.4)		2 (50.0)	37 (72.5)	
Unknown	1 (1.1)	5 (9.1)		0 (0.0)	1 (1.7)		0 (0.0)	5 (9.8)	

Abbreviations: CC, positive cancer cachexia status; Non-CC, negative cancer cachexia status; diff., differentiation; CA19-9, carbohydrate antigen 19-9; BMI, body mass index; CRP, C-reactive protein. The bold value indicates *p* < 0.05.

**Table 2 jcm-07-00502-t002:** The clinicopathologic characteristics and expression of IL-8 in pancreatic tumor specimens.

Characteristic	Tissue Expression (%)		
	Low (*n* = 52)	High (*n* = 39)	*p*-Value
**Sex**			
Male	33 (63.5)	26 (66.7)	0.751
Female	19 (36.5)	13 (33.3)	
**Tumor Location**			
Head	32 (61.5)	24 (61.5)	0.789
Neck	4 (7.7)	2 (5.1)	
Body/tail	10 (19.2)	6 (15.4)	
Uncinate process	6 (11.5)	7 (17.9)	
unknown	0 (0.0)	0 (0.0)	
**Tumor Size**			
<3 cm	24 (46.2)	10 (25.6)	**0.045**
≥3 cm	28 (53.8)	29 (74.4)	
**Lymph Nodes**			
Negative	25 (48.1)	21 (53.8)	0.586
Positive	27 (51.9)	18 (46.2)	
**Margin Status**			
R0	37 (71.2)	28 (71.8)	0.747
R1	12 (23.1)	10 (25.6)	
R2	3 (5.8)	1 (2.6)	
**Tumor Grade**			
Poorly diff.	9 (17.3)	8 (20.5)	0.542
Moderate diff.	30 (57.7)	18 (46.2)	
Well diff.	13 (25.0)	13 (33.3)	
**Stage**			
I	7 (13.5)	4 (10.3)	0.956
II	42 (80.8)	32 (82.1)	
III	2 (3.8)	2 (5.1)	
IV	1 (1.9)	1 (2.6)	
**CA19-9**			
<37 U/mL	15 (28.8)	4 (10.3)	**0.031**
>37 U/mL	37 (71.2)	35 (89.7)	
**Adjuvant Therapy**			
Yes	18 (34.6)	18 (46.2)	0.265
No	34 (65.4)	21 (53.8)	
**Diabetes Mellitus**			
Yes	20 (38.5)	8 (20.5)	0.066
No	32 (61.5)	31 (79.5)	
**BMI**			
<20 kg/m^2^	26 (50.0)	27 (71.1)	0.120
>20 kg/m^2^	22(42.3)	10 (26.3)	
unknown	4 (7.7)	1 (2.6)	
**Fatigue**			
Yes	20 (38.5)	17 (43.6)	0.622
No	32 (61.5)	22 (56.4)	
**Anorexia**			
Yes	15 (28.8)	23 (59.0)	**0.004**
No	37 (71.2)	16 (41.0)	
**CRP**			
<5 mg/L	8 (15.4)	8 (20.5)	0.307
>5 mg/L	43 (82.7)	28 (71.8)	
unknown	1 (1.9)	3 (3.3)	
**Hemoglobin**			
<12 g/dL	20 (38.5)	16 (41.0)	0.804
>12 g/dL	32 (61.5)	23 (59.0)	
unknown	0 (0.0)	0 (0.0)	
**Albumin**			
<3.2 g/dL	10 (19.2)	8 (20.5)	0.498
>3.2 g/dL	42 (80.8)	30 (76.9)	
unknown	0 (0.0)	1 (2.6)	

Abbreviations: diff, differentiation; CA19-9, carbohydrate antigen 19-9; BMI, Body mass index; CRP, C-reactive protein.The bold value indicates *p* < 0.05.

**Table 3 jcm-07-00502-t003:** Prognostic factors associated with survival in the multivariate analysis.

Variable	No. of Patients	Overall Survival			Disease-Free Survival		
		HR	95% CI	*p*-Value	HR	95% CI	*p*-Value
**Cancer Cachexia**							
No	36	1	5.390–64.382	**0.000**	1	1.092–2.685	**0.019**
Yes	110	18.629			1.712		
**Weight Loss**							
<5%	40	1	2.617–22.544	**0.000**	1	1.027–2.433	**0.038**
>5%	106	7.682			1.581		
**BMI**							
≤20 kg/m^2^	86	1	0.113–1.037	0.058	1	0.766–1.748	0.489
>20 kg/m^2^	54	0.342			1.157		
Unknown	6						
**Fatigue**							
No	74	1	0.829–2.058	0.250	1	0.877–1.839	0.207
Yes	72	1.306			1.269		
**Anorexia**							
No	74	1	1.359–3.757	**0.002**	1	0.959–2.065	0.081
Yes	72	2.260			1.407		
**CRP**							
<5 mg/L	22	1	0.989–3.435	0.054	1	0.645–1.717	0.837
>5 mg/L	114	1.843			1.053		
Unknown	10						
**Hemoglobin**							
<12 g/dL	60	1	0424–1.236	0.236	1	0.081–1.612	0.182
>12 g/dL	84	0.724			0.362		
Unknown	2						
**Albumin**							
<3.2 g/dL	29	1	0.404–1.138	0.141	1	0.179–1.388	0.183
>3.2 g/dL	111	0.678			0.498		
Unknown	6						
**IL-8 expression**							
Low	50	1	0.825–2.742	0.183	1	1.080–2.843	0.023
High	41	1.504			1.752		
**Serum IL-8**							
Low	55	1	1.045–2.495	**0.031**	1	1.021–2.319	**0.040**
High	55	1.615			1.539		
**CC and IL-8 expression**							
Group 1	20	1	Reference		1	Reference	
Group 2	42	1.742	0.727–4.170	0.213	1.508	0.784–2.902	0.219
Group 3	29	2.555	1.042–6.261	**0.040**	1.996	1.008–3.954	**0.047**
**CC and Serum IL-8**							
Group 1	13	1	Reference		1	Reference	
Group 2	45	2.911	1.117–7.585	**0.029**	1.423	0.723–2.821	0.307
Group 3	52	3.840	1.508–9.777	**0.005**	1.928	0.990–3.755	**0.045**

Abbreviations: HR, hazard ratio; CI, confidence interval; BMI, body mass index; CRP, C-reactive protein; CC, cancer cachexia; IL-8, interleukin-8. CC and IL-8 expression: Group 1, without CC and low IL-8; Group 2, without CC and high IL-8 or CC and low IL-8; Group 3, CC and high IL-8. CC and Serum IL-8: Group 1, without CC and low Serum IL-8; Group 2, without CC and high Serum IL-8 or CC and low Serum IL-8; Group 3, CC and high Serum IL-8. The bold value indicates *p* < 0.05.

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
