# Peer review of "Elevated Serum Interleukin-8 Level Correlates with Cancer-Related Cachexia and Sarcopenia: An Indicator for Pancreatic Cancer Outcomes"

_jcm, 2018, doi:10.3390/jcm7120502_

Reviewer 1 Report

Dear authors, 

you address an important question and issue in pancreatic cancer, namely cachexia, which might contribute to bad outcomes in this highly lethal cancer. However, I believe that this manuscript - although nicely written - has a few serious technical flaws as outlined below. It is not grouped in major/minor issues, but listed in the order of the manuscript:

Fig. 1: arrows going up and down, this is very confusing for the reader; please simplify.

Fig. 2: A) p=0.000 does not exist, please add real numbers here

In Section 3.1 and Table 1, what do you exactly mean with survival? What is the difference in your definition of CC and Weight loss? What is the range of CA19-9 and would it make sense to look at CA19-9+ patients and then set a threshold, e.g. 500? Also, I don't think it is useful to compare 4 non-CC to 51 CC patients in the locally advanced group; the latter applies to the whole manuscript.

Fig. 4: To me, the big difference seems to stem from the fact that IL6 is a lot higher in locally advanced cancers, which are pretty much all CC and that has a significant influence on the graph of all patients. Also, I wonder what distinguishes the outliers of Fig. 4E CC.

A similar phenomenon can be observed for IL-8.

Moreover, all graphs of one row (A-C, D-F, etc.) should have the same scaling on the y-axis; in other words, don't show IL1b in A and C with 200-400-600, whereas B goes in steps of 20 from 80 to 180! The same applies to H and K; same issue in Fig. 6.

Fig. 8: why did you only measure 10 samples of adjacent normal pancreatic tissue vs. 91 tumor samples? This is at odds. How many view fields per core were evaluated?

Line 238: rephrase 'annoying'

Reviewer 2 Report

The study is well done and clearly presented. 

The authors need to provide legend for figure 1. Why have dotted or dashed lines?

Authors should elaborate on why they looked at these cytokines and not others. Are these cytokines the ones usually different in tumor models? What does changes in these cytokines signify? 

In their study is there a different between males and females? 

Author Response

Round  2

Reviewer 1 Report

I believe that the questions raised by the reviewers have been answered satisfactorily.

J. Clin. Med. EISSN 2077-0383 Published by MDPI AG, Basel, Switzerland RSS E-Mail Table of Contents Alert
Back to Top